# Mitochondrial Dysfunction and the Glycolytic Switch Induced by Caveolin-1 Phosphorylation Promote Cancer Cell Migration, Invasion, and Metastasis

**DOI:** 10.3390/cancers14122862

**Published:** 2022-06-10

**Authors:** Natalia Díaz-Valdivia, Layla Simón, Jorge Díaz, Samuel Martinez-Meza, Pamela Contreras, Renato Burgos-Ravanal, Viviana I. Pérez, Balz Frei, Lisette Leyton, Andrew F. G. Quest

**Affiliations:** 1Cellular Communication Laboratory, Center for Studies on Exercise, Metabolism and Cancer (CEMC), Program of Cell and Molecular Biology, Institute of Biomedical Sciences (ICBM), Faculty of Medicine, Universidad de Chile, Santiago 8380000, Chile; ndiazv@ciq.uchile.cl (N.D.-V.); lsimonujam@gmail.com (L.S.); jorgediazfuentes1@gmail.com (J.D.); samuel.martinez.meza@gmail.com (S.M.-M.); orellana.paz@gmail.com (P.C.); raburgos@uc.cl (R.B.-R.); 2Advanced Center for Chronic Diseases (ACCDiS), Faculty of Medicine, Universidad de Chile, Santiago 8380000, Chile; 3Linus Pauling Institute, Department of Biochemistry and Biophysics, Oregon State University, Corvallis, OR 97331, USA; viviana.perez@oregonstate.edu (V.I.P.); balz.frei@oregonstate.edu (B.F.)

**Keywords:** caveolin-1, metabolic switch, mitochondrial complex IV, tyrosine-14 phosphorylation, PTP1B, metastasis

## Abstract

**Simple Summary:**

Caveolin-1 (CAV1) is a membrane protein that has been attributed a dual role in cancer, acting at early stages as a tumor suppressor and in later stages of the disease as a promoter of metastasis. In the latter case, enhanced expression of CAV1 favors the malignant phenotype and correlates with a poorer prognosis of the patients. Bearing in mind that the reprogramming of energy metabolism is required in cancer cells to meet both the bioenergetic and biosynthetic needs to sustain increased proliferation, migration, and invasion, we evaluated the metabolism of metastatic cells expressing or not CAV1. In this study, we show that the expression of CAV1 promotes in cancer cells a metabolic switch to an aerobic, glycolytic phenotype by blocking mitochondrial respiration.

**Abstract:**

Cancer cells often display impaired mitochondrial function, reduced oxidative phosphorylation, and augmented aerobic glycolysis (Warburg effect) to fulfill their bioenergetic and biosynthetic needs. Caveolin-1 (CAV1) is a scaffolding protein that promotes cancer cell migration, invasion, and metastasis in a manner dependent on CAV1 phosphorylation on tyrosine-14 (pY14). Here, we show that CAV1 expression increased glycolysis rates, while mitochondrial respiration was reduced by inhibition of the mitochondrial complex IV. These effects correlated with increased reactive oxygen species (ROS) levels that favored CAV1-induced migration and invasion. Interestingly, pY14-CAV1 promoted the metabolic switch associated with increased migration/invasion and augmented ROS-inhibited PTP1B, a phosphatase that controls pY14 levels. Finally, the glycolysis inhibitor 2-deoxy-D-glucose reduced CAV1-enhanced migration in vitro and metastasis in vivo of murine melanoma cells. In conclusion, CAV1 promotes the Warburg effect and ROS production, which inhibits PTP1B to augment CAV1 phosphorylation on tyrosine-14, thereby increasing the metastatic potential of cancer cells.

## 1. Introduction

Chronic uncontrolled proliferation is one of the hallmarks of cancer cells [1]. To accomplish this, the cell metabolism needs to be reprogrammed to provide the necessary metabolic intermediates for protein, nucleic acid, and lipid synthesis [2]. In aerobic conditions, normal cells metabolize glucose to pyruvate by glycolysis in the cytosol and later to CO_2_ in the mitochondria; however, cancer cells frequently convert to a mainly glycolytic metabolism even in the presence of oxygen, a phenomenon known as the Warburg effect [3,4]. Such metabolic reprograming is rather inefficient, in that cells must compensate reduced efficiency in ATP production by overexpression of glucose transporters, like GLUT1, and increased glucose uptake [5,6]. However, increased glycolysis allows the cancer cells to use glucose, the most abundant nutrient present, to produce large amounts of ATP, which favors more rapid cell migration and invasion. Moreover, and rather importantly, glucose catabolism provides the cells with metabolic intermediates required for the synthesis of biomolecules essential for proliferation [7]. Therefore, the Warburg effect satisfies both the bioenergetic and biosynthetic needs of cancer cells. As a consequence of this metabolic switch, cancer cells increase the production of lactic acid [8], which can induce death in normal cells [9] and the degradation of the extracellular matrix, to promote migration and invasion of the cancer cells [7,10].

Caveolin-1 (CAV1), a membrane-associated scaffolding protein with a dual role in tumor development and progression, has been implicated in the metabolic reprogramming of cells in the tumor environment. More specifically, decreased CAV1 expression in carcinoma-associated fibroblasts (CAFs) is sufficient to increase the intracellular levels of reactive oxygen species (ROS), promote autophagy and glycolysis, and diminish mitochondrial respiratory metabolism, thereby favoring tumor progression [11,12,13,14]. These examples emphasize the importance of CAV1 expression in regulating the metabolism of cells in the tumor stroma environment and how this can affect cancer cell behavior.

Elevated expression of CAV1 in colon cancer cells increases glucose uptake and ATP production by stimulating transcription of the glucose transporter 3. Moreover, the depletion of CAV1 leads to AMPK activation followed by a p53-dependent G_1_ cell-cycle arrest, autophagy, reduced glucose uptake and intracellular ATP levels, together with lactate accumulation. This evidence suggests that CAV1 expression in these cancer cells favors aerobic glycolysis [15]. CAV1 also interacts with the low-density lipoprotein receptor protein 6 (LRP6) and stimulates the kinase activities of the insulin and IGF-I receptors (IGF-IR/IR) in PC-3 cells, as well as in primary human prostate cancer tumors and metastases. In this manner, LRP6 and CAV1 stimulate aerobic glycolysis by increasing the expression of glycolytic enzymes in prostate cancer cells [16].

CAV1 is subject to posttranslational modifications, such as phosphorylation on tyrosine 14 (Y14) by the non-receptor protein tyrosine kinases Src, Fyn, and Abl. This occurs in response to different stimuli like insulin, ultraviolet radiation, hydrogen peroxide, hyperosmolarity, and shear stress [17,18,19,20,21,22,23,24,25,26]. Metastatic breast cancer cells express high levels of CAV1 as well as elevated levels of Y14 phosphorylation, which are associated with increased cell migration by promoting focal adhesion turnover, polarization, persistency, speed, and directionality of migration [27,28,29,30,31]. All these effects of CAV1 are blocked by the pharmacological inhibition of phosphorylation on Y14 or by introducing a non-phosphorylatable caveolin-1 (Y14F) mutation [31]. Alternatively, phosphatases, such as the non-receptor protein tyrosine phosphatase 14 (PTPN14) [32] and type 1B (PTP1B) [33,34] reportedly dephosphorylate CAV1 on tyrosine 14.

Bearing this in mind, the aim of this study was to evaluate the effect of CAV1 expression on the metabolism of metastatic cancer cells lacking E-cadherin and how Tyr-14 phosphorylation participates in these events. Our results demonstrate that CAV1 expression in metastatic cells increases glycolysis and blocks the mitochondrial complex IV, which increases ROS levels that inhibit PTP1B, thereby augmenting phosphorylation of CAV1 on Tyr-14 to promote cell migration, invasion, and metastasis.

## 2. Materials and Methods

### 2.1. Materials

The rabbit polyclonal anti-CAV1, mouse monoclonal anti-CAV1, mouse monoclonal anti-pY14-CAV1 and mouse monoclonal anti-PTP1B antibodies were from BD Transduction Laboratories, San Jose, CA, USA. Goat anti-rabbit and goat anti-mouse IgG antibodies coupled to horseradish peroxidase (HRP) were from Merck-Millipore (Burlington, MA, USA) and KPL Laboratories (Gaithersburg, MD, USA), respectively. EZ-ECL chemiluminescent substrate and the BCA protein determination kit were purchased from Thermo Fisher Scientific (Waltham, MA, USA). The Plasmid Midi Kit was from Qiagen (Hilden, Germany). Human fibronectin was obtained from Becton Dickinson (San Jose, CA, USA) and Hygromycin was from Calbiochem (San Diego, CA, USA). Fetal bovine serum (FBS) was purchased from Biological Industries, Israel. Cell culture media and antibiotics were from Invitrogen (Carlsbad, CA, USA). Oligomycin, Carbonyl cyanide-*p*-trifluoromethoxyphenylhydrazone (FCCP), Rotenone, Glucose, 2-Deoxyglucose (2-DG), and Antimycin A were from Seahorse Bioscience, Agilent Technologies (Santa Clara, CA, USA). Ascorbate, Succinate, Pyruvate, Mito-TEMPO, and N-acetylcysteine were from Sigma-Aldrich (Saint Louis, MO, USA). Protein A/G sepharose and CinnGEL 2-methylester (CinnGEL 2Me) were from Santa Cruz Biotechnology (Santa Cruz, CA, USA). The L-Lactate assay kit was from Eton Bioscience Inc. (San Diego, CA, USA) and 2-DG from Sigma-Aldrich (Saint Louis, MO, USA).

### 2.2. Cell Culture

The metastatic murine melanoma cells B16F10 (ATCC, #CRL6475, provided by Laurence Zitvogel, Institut Gustave Roussy, Villejuif, France) were maintained in RPMI 1640 medium. The colon cancer cell line HT29(US), a metastatic derivative of HT29(ATCC) cells from ATCC (ATCC HTB-38) that we have employed in previous studies [35,36,37], was cultured in high-glucose DMEM. Breast cancer MDA-MB-231 cells were cultured in DMEM-F12 medium. The media were supplemented with 10% FBS and 100 U/mL of penicillin together with 100 μg/mL of streptomycin sulfate, and cells were cultured at 37 °C in a humidified atmosphere containing 5% CO_2_.

Stable transfection with the plasmids pLacIOP (referred to as Mock) and pLacIOP-caveolin-1 (referred to as CAV1, which contains the full-length dog caveolin-1 sequence, NCBI Reference Sequence: NP_001003296.1) allows the IPTG-inducible CAV1 expression in HT-29(US) and B16-F10 cells. In the case of MDA-MB-231 cells, endogenous CAV1 was stably knocked down as described previously [31,35,36]. Control MDA-MB-231 cells were infected with a lentivirus encoding a nonspecific shRNA sequence (plasmid 1864; Addgene, Cambridge, MA, USA). Transduced MDA-MB-231 cell lines were selected and maintained in culture medium containing 2 mg/mL puromycin. B16F10 cells expressing mutations of CAV1, namely, B16F10(CAV1/Y14F) and B16F10(CAV1/Y14E) cells were described previously [38]. Cell lines stably transfected with pLacIOP controls (Mock cells) or CAV1 constructs were selected in culture medium containing 800 μg/mL hygromycin. 

### 2.3. Oxygen Consumption 

2 × 10^5^ B16F10 (Mock, CAV1), 1 × 10^5^ HT29(US) (Mock and CAV1) or 4 × 10^5^ MDA-MB-231 (shCAV1 o shC) cells were seeded in 24 well plates (XF24). After 24 h, the cells were incubated for 1 h in 1 mL of Seahorse Bioscience XF24 Calibrant pH 7.4 supplemented with 100 mM pyruvate and 5.5 mM glucose at 37 °C. Every 10 min, four measurements were made to determine the basal respiration. Subsequently, the cells were treated with either oligomycin (1 μM), an inhibitor of ATP synthesis, which permits determining the percentage of oxygen consumption dedicated to ATP synthesis and the percentage of oxygen that is required to overcome the leakage of protons through the inner mitochondrial membrane. Then, carbonyl cyanide-*p*-trifluoromethoxyphenylhydrazone (FCCP, 0.5 μM), a mitochondrial respiratory chain uncoupling agent, was injected. This results in the collapse of the mitochondrial membrane potential, leading to rapid consumption of energy and oxygen without generating ATP. The difference in maximum oxygen consumption after treatment with FCCP and basal oxygen consumption is equivalent to the reserve respiratory capacity. The ability of cells to respond to stress under conditions of high energy demand is influenced by the bioenergetic capacity of the mitochondria, which reflects the integrity of the electron transport chain. Finally, the cells were treated with Rotenone (100 nM), an inhibitor of complex I, which prevents the conversion of the potential energy as NADH to useful energy in the form of ATP, decreasing oxygen consumption, allowing the determination of non-mitochondrial respiration. After each treatment, four measurements were made every 10 min. Oxygen consumption was determined by changes in the emission of an oxygen-sensitive fluorophore embedded in the sensor of the XF24 Seahorse extracellular flow analyzer (Seahorse Bioscience, Agilent Technologies).

### 2.4. Glycolytic Capacity 

2 × 10^5^ B16F10 (Mock, CAV1, Y14F and Y14E), 1 × 10^5^ HT29(US) (Mock and CAV1), or 4 × 10^5^ MDA-MB-231 (shCAV1 o shC) cells were seeded in 24 well plates (XF24). After 24 h, the cells were washed twice with 300 μL of the working media (DMEM supplemented with NaCl (143 mM), L-glutamine (2 mM) and glucose (10 mM), pH 7.35), and then 225 μL of the working media was added to each well, and measurements were made every 10 min to determine the baseline extracellular acidification rate (ECAR). Later, the cells were treated with saturating concentrations of glucose (10 mM), which produced a rapid increase in the ECAR. This glucose-induced response is a measure of the glycolytic flow under basal conditions. Subsequently, the cells were treated with Oligomycin (1 μM), which inhibits mitochondrial ATP production, thereby augmenting energy production due to glycolysis and increasing ECAR. This permitted determining the maximum glycolytic capacity of the cells. Finally, the cells were treated with the glucose analog, 2-Deoxy-D-glucose (2-DG, 100 mM), which inhibits glycolysis by competitive binding to hexokinase, the first enzyme in the glycolytic pathway, producing a decrease in the ECAR. This fact confirms that the ECAR produced in this experiment is due to glycolysis. After each treatment, three measurements were made every 10 min. 

Acidification of the extracellular media was determined as an indicator of cellular glycolytic activity by measuring changes in the emission of a proton sensitive fluorophore embedded in the sensor of XF24 Seahorse extracellular flow analyzer (Seahorse Bioscience, Agilent Technologies).

### 2.5. Mitochondrial Respiratory Activity

2 × 10^5^ B16F10 (Mock or CAV1) cells were seeded in 24 well plates (XF24), and after 24 h the cells were permeabilized using 1 nM of XF Plasma Membrane Permeabilizer (XF PMP, Seahorse Bioscience, Agilent Technologies) plus ADP 4 mM in MAS 1X media (see Table 1), supplemented with pyruvate (10 mM), as a mitochondrial complex I substrate. Immediately after the incubation with XF PMP, oxygen consumption was measured three times every 4 min. Later, the cells were treated with rotenone (2 μM), succinate (10 mM), antimycin A (2 μM), and ascorbate (10 mM) plus TMPD (100 μM). After each treatment, three measurements were made every 4 min. Oxygen consumption was determined by measuring changes in the emission of an oxygen-sensitive fluorophore embedded in the sensor of XF24 Seahorse extracellular flow analyzer (Seahorse Bioscience, Agilent Technologies).

### 2.6. ROS Determination by Flow Cytometry

B16F10 (Mock or CAV1) cells were transfected using lipofectamine with the hydrogen peroxide sensors (HyPer), which accumulate in mitochondria (pHyPer-dMito) or the cytosol (pHyPer-cyto) and are excitable at 420 and 500 nm, respectively, and emit at 516 nm. Cells were analyzed by flow cytometry in a FACS CantoA (Becton Dickinson, San Jose, CA, USA).

### 2.7. Migration and Invasion Assays 

Cell migration was evaluated using Boyden Chamber assays (Transwell Costar, 6.5-mm diameter, 8-mm pore size). The inserts were incubated at 37 °C for 2 h (B16F10 and MDA-MB-231 cells) or 5 h (HT29(US)), whereas invasion was evaluated in Matrigel assays (BD Biosciences 354480, San Jose, CA, USA ) after 24 h, as we described previously [37].

### 2.8. Lactate Production 

6 × 10^5^ cells were seeded in 6 cm plates, and 24 h later B16F10 (Mock or CAV1) and HT29(US) (Mock or CAV1) cells were treated with IPTG 1 mM for 48 or 24 h, respectively, or MDA-MB-231 (shCAV1 o shC). After this time, culture media were collected and diluted in PBS (1:2, 1:4, and 1:8), and cells were left untreated for 24 h. Absorbance was measured at 490 nm according to the instructions provided by the manufacturer (Eton Bioscience Inc.) of the L-Lactate assay kit.

### 2.9. Western Blotting

Protein extracts (50 μg) were separated by SDS-PAGE), transferred to nitrocellulose, blocked in PBS containing 5% non-fat milk, and probed overnight at 4 °C with the first antibodies diluted in PBS 1% Tween-20. Protein loading in each lane was assessed with an anti-β-actin antibody (1:5000). Goat anti-rabbit IgG antibodies coupled to horseradish peroxidase were used as secondary antibodies, and the signal was developed by EZ-ECL. Protein bands were quantified by densitometric analysis using the ImageJ 1.8.0_112a software (available at https://imagej.nih.gov). All the uncropped western blots are available in the Appendix A.

### 2.10. PTP1B Activity 

B16F10 (Mock or CAV1) cells were harvested in ice cold PBS, supplemented with protease and phosphatase inhibitors (PMSF 1 mM, Na3VO4 1 mM, NaF 10 mM, Benzamide 100 μg/mL, Antipain 10 μg/mL, and Leupeptin 12.5 μg/mL). Cells were lysed for 15 min in 20 mM Tris (pH 7.4), 150 mM NaCl, 1% NP40, and the cocktail of protease and phosphatase inhibitors. At least 2 mg of proteins were immunoprecipitated using 2.5 μg of primary antibody for 3 h at 4 °C on a rotating shaker. Then, a protein A/G sepharose mix (50 μL, 1:1) was added, and samples were incubated for 12 h on a rotating shaker at 4 °C. To measure phosphatase activity, p-nitrophenyl phosphate (5.5 μL) was used as a substrate. The mix was incubated at 37 °C, and absorbance was measure at 415 nm at different time points in a spectrophotometer as described [34].

### 2.11. In Vivo Experimental Metastasis Assay

2 × 10^5^ B16F10(Mock) or B16F10(CAV1) cells treated for 24 h with 2-Deoxy-D-glucose (2-DG, 1 mM) were injected intravenously into the tail vein of C57BL/6 mice of 8–12 weeks of age. On day 21 post-injection, the mice were sacrificed. Lungs were fixed in Fekete’s solution (95% ethanol, 37% formaldehyde, and glacial acetic acid in water). Black metastatic tissue was separated from the rest of the lung and weighed. Metastasis was expressed as black tumor mass/total lung mass in percent post-fixation. 

### 2.12. Statistical Analysis 

Data are expressed as the mean ± standard error of mean (SEM) of results from three independent experiments and were analyzed using the non-parametric Kruskal–Wallis test for multiple comparisons with a post test of Dunn. Significance (*p*-value) was set at *p* < 0.05 or less. All data were processed using Prism 9.0c. (GraphPad Software, San Diego, CA, USA, http://www.graphpad.com).

## 3. Results

### 3.1. Caveolin-1 Expression Decreases Oxygen Consumption and Increases the Glycolysis Rate in Metastatic Cancer Cell Lines 

Metabolic reprogramming is one of the acquired characteristics of cancer cells during tumor progression. We have previously reported on how the expression of CAV1 and its phosphorylation on tyrosine-14 (pY14-CAV1) promote migration, invasion, and metastasis of colon and breast cancer, as well as melanoma cells lacking E-cadherin [31,38,39]. Given this ability of CAV1, and the aforementioned evidence linking CAV1 expression to metabolic changes, we sought to determine whether CAV1 and particularly Y14 phosphorylation might affect mitochondrial metabolism. To that end, we analyzed the metabolic state of metastatic cells and how this was altered by the expression of CAV1. For that purpose, we used B16F10 mouse melanoma and HT29(US) colon cancer cells transfected with the IPTG-inducible plasmid pLacIOP (Mock) or pLacIOP-caveolin-1 (CAV1) vectors [35,38]. Alternatively, MDA-MB-231 human breast cancer cells that express high endogenous CAV1 levels were transduced with an shRNA to knock down CAV1 (shCav1) or with a control shRNA (shScr) [31]. The rate of oxygen consumption (OCR) was determined as an indicator of the mitochondrial respiration and the extracellular acidification rate (ECAR) as an indicator of the glycolytic rate in cells.

Cells that do not express CAV1 (Mock and shCAV1) have a higher basal OCR than cells that express CAV1. In B16F10 and MDA-MB-231 cells that express CAV1, the amount of oxygen used for ATP synthesis and the reserve respiratory capacity were decreased compared to cells that do not express CAV1 (Figure 1A,B). For HT29(US) colon cancer cells, the expression of CAV1 also resulted in reduced basal respiration and lower respiratory capacity, although the difference was not as pronounced (Figure 1C). In all three cell lines, the non-mitochondrial respiration, estimated by the addition of Rotenone, was not significantly altered by the expression of CAV1 (Figure 1A–C). Together these results indicated that the expression of CAV1 in metastatic cells generates a mitochondrial impediment, which results in reduced mitochondrial respiration and oxygen consumption.

Because the expression of CAV1 decreases the OCR in metastatic cells (Figure 1A–C), we then evaluated whether the expression of CAV1 induced metabolic reprogramming to an aerobic glycolytic metabolism, better known as the Warburg effect. For this, we determined the extracellular acidification rate (ECAR) in an extracellular flow analyzer, as a measure of the glycolytic rate of the B16F10, MDA-MB-231, and HT29(US) cells that express or not CAV1. Acidification is attributable to the release of protons to the extracellular medium linked to lactate production due to glucose catabolism. 

As mentioned, no differences in basal glycolysis were observed for B16F10 (Figure 1D) and MDA-MB-231 (Figure 1E) cells expressing (CAV1, shScr, red line) or not CAV1 (Mock, shCAV1, blue line). However, in HT29(US) cells (Figure 1F), the expression of CAV1 resulted in an increase in basal glycolysis. After treatment with a saturating concentration of glucose, the glycolytic flux increased in all three cell lines expressing or not CAV1; however, the increase was more significant in cells expressing CAV1 (Figure 1D–F). The same was also the case for the maximum glycolytic capacity (Figure 1D–F). Thus, the expression of CAV1 in these metastatic cells increased glycolysis under aerobic conditions.

### 3.2. Caveolin-1 Expression Diminishes Oxygen Consumption by Blocking the Mitochondrial Complex IV in B16F10 Cells

Since CAV1 expression decreased the mitochondrial reserve respiratory capacity (Figure 1A–C), we wished also to determine if the activity of the mitochondrial complexes was affected. To this end, B16F10 cells that express (B16F10(CAV1)) or not CAV1 (B16F10(Mock)) were transiently permeabilized and maintained in the presence of pyruvate, as a complex I substrate, and oxygen consumption was measured. Subsequently, the cells were treated with the complex I inhibitor Rotenone, which blocked NADH-associated mitochondrial respiration. Then, cells were treated with Succinate, which leads to consumption of the electrons that entered the ubiquinone pool via the succinate dehydrogenase complex II, thus avoiding the inhibition of complex I. Ultimately, cells were treated with Antimycin A, an inhibitor of complex III, and this inhibition was bypassed by the addition of ascorbate and TMPD 100, which deliver electrons to and reduce the cytochrome C oxidase, a substrate of complex IV (Figure 2A). 

The cells that did not express CAV1 (Mock, blue lines, Figure 2B) were able to circumvent the inhibition of each mitochondrial complex, in the presence of the appropriate substrates, and continue with the mitochondrial respiration. However, for CAV1-expressing cells (red line, Figure 2B), oxygen consumption was diminished (see Figure 1), and mitochondrial respiration was not restored even after bypassing complex III inhibition with Antimycin A. This suggests that CAV1 either blocks the mitochondrial complex IV or renders cells unable to recover from the increased reactive oxygen species produced by complex III, which may explain why the CAV1-expressing cells consumed less oxygen and switched to aerobic glycolysis. 

### 3.3. Caveolin-1 Expression Increased Cell Migration and Invasion in a Manner Dependent on ROS Production

The block of the mitochondrial complex IV or the inability to recover from Antimycin A treatment induced by the expression of CAV1 was predicted to translate into the accumulation of intracellular ROS. To test this, B16F10 cells expressing (B16F10(CAV1)) or not CAV1 (B16F10(Mock)) were transfected with a fluorescent hydrogen peroxide sensor plasmid (HyPer) that either has a mitochondrial destination signal (pHyPer-dMito) or remains in the cytoplasm (pHyPer-cyto). These sensors permit determining by flow cytometry where in cells ROS levels increase (Figure 3A). The expression of CAV1 induced an increase in mitochondrial (Figure 3A, black bars) and cytoplasmic ROS (Figure 3A, striped bars). We then determined whether ROS accumulation was responsible for the increase in cell migration and invasion due to CAV1 reported in previous studies by our group [31,32,38]. To that end, B16F10(Mock) and (CAV1) cells were pre-treated with the generic antioxidant N-Acetylcysteine (NAC, Figure 3B) or with the mitochondria-targeted anti-oxidant Mito-Tempo (Figure 3C). The treatment with either one of the two antioxidants sufficed to block CAV1-enhanced cell migration and invasion. These observations can be taken to suggest that augmented intracellular ROS due to CAV1 expression is responsible for the increase in cell migration and invasion. 

### 3.4. Phosphorylation on Tyrosine 14 Is Necessary for Maintenance of the Metabolic Phenotype Induced by the Expression of Caveolin-1

High levels of CAV1 expression and phosphorylation on Y14 were observed and associated with increased cell migration by promoting focal adhesion turnover, polarization, persistency, speed, and directionality of migration [31]. All these functions of CAV1 were blocked by introducing a non-phosphorylatable CAV1(Y14F) mutation (Figure 4). Indeed, CAV1-induced migration (Figure 4A) and invasion (Figure 4B) were not observed upon CAV1(Y14F) expression. Therefore, the metastasis-promoting role of caveolin-1 depends on Y14 phosphorylation.

Interestingly, the presence of CAV1 wild type and phosphomimetic CAV1(Y14E) increased lactate production (Figure 4C). These results demonstrate that CAV1 phosphorylation on tyrosine 14 is required for CAV1 to promote migration and invasion, as well as for the CAV1-induced metabolic switch.

### 3.5. Caveolin-1-Increased Lactate Release Is Inhibited by the Treatment of Metastatic Cancer Cells with Antioxidants

Cancer cells alter their metabolism to promote growth, survival, and proliferation. A characteristic of such alterations in metabolism is the increased glucose uptake and its conversion to lactate. Previous reports showed that CAV1 depletion down-regulated lactate accumulation in human colon [15] and prostate [16] cancer cell lines. These observations led us to ask whether the expression of CAV1 modulated lactate release. To test this possibility, the lactate released to the media by cells with low-CAV1 levels (mouse melanoma B16F10(Mock), human breast cancer MDA-MB-231(shCAV1), and human colon adenocarcinoma HT29(US)(Mock)) was compared with the lactate levels in media from cells with high CAV1 expression (B16F10(CAV1), MDA-MB-231(shC), and HT29(US)(CAV1)). For the MDA-MB-231, cancer cells which express CAV1 endogenously, CAV1 levels were reduced using a specific short hairpin construct. We observed that the presence of CAV1 in these cells increased lactate accumulation in the media more than three-fold for B16F10 and HT29(US) cells and more than two-fold for MDA-MB-231 cells (Figure 5A).

To test the relevance of ROS formation in mitochondria and in the cytosol, we used the two aforementioned anti-oxidant molecules. Mito-TEMPO is a mitochondrially targeted antioxidant and a specific scavenger of mitochondrial superoxide that easily passes through lipid bilayers and accumulates several hundred-fold in mitochondria. Alternatively, N-acetylcysteine (NAC) is an antioxidant precursor that increases cytosolic pools of free radical scavengers. Considering that CAV1 phosphorylation on tyrosine-14 (pY14) is important for metastatic cell behavior, we also evaluated how the expression of non-phosphorylatable CAV1(Y14F) or phosphomimetic CAV1(Y14E) in B16F10 cells affected lactate accumulation in the culture media. 

For B16F10 cells, Mito-TEMPO had no effect on lactate levels in the media from cells with low CAV1 levels (Mock). In the presence of non-phosphorylatable CAV1(Y14F), Mito-TEMPO treatment also had no significant effect on media lactate levels, but for CAV1-expressing cells, lactate levels were reduced by one third and for CAV1(Y14E) cells by 50%, as compared with non-treated (NT) cells (Figure 5B, white versus black bars). Similarly, for HT29(US) cells with low CAV1 levels (Mock), the media lactate levels were not reduced by Mito-Tempo, while in cells expressing high CAV1 levels, lactate accumulation in the media was reduced by 70% compared with non-treated (CAV1) cells (Figure 5B, white versus black bars). Finally, in MDA-MB-231 cells, Mito-Tempo treatment of cells with low CAV1 levels (shCAV1) had no effect on media lactate levels, but for cells with high CAV1 levels (shC), the media lactate levels were reduced by 50% (Figure 5B, white versus black bars).

On the other hand, NAC reduced at least ten-fold the media lactate levels in a manner independent of CAV1 (mock and CAV1) or CAV1 phosphorylation (Y14F and Y14E) in B16F10 cells (Figure 5B, white versus gray bars). Similar results were obtained with HT29(US) cells. Media lactate levels for cells expressing low levels of CAV1 were reduced by 50%, while for cells with high CAV1 levels the media lactate levels were reduced more than twenty-fold compared with non-treated cells (Figure 5B, white versus gray bars). Moreover, in MDA-MB-231 cells with low CAV1 expression (shCAV1), NAC treatment had no effect on media lactate levels, but in cells with high CAV1 levels (shC), the media lactate levels were reduced ten-fold (Figure 5B, white versus gray bars). In summary, these results can be taken to suggest that the expression and phosphorylation of CAV1 induced ROS accumulation and the production of lactate by triggering mitochondrial dysfunction. 

### 3.6. Inhibition of PTP1B Activity Enhances Cell Migration, Invasion, and Lactate Production by Increasing CAV1 Phosphorylation

CAV1 phosphorylation on tyrosine-14 is not only controlled by kinases that increase phosphorylation levels but also by phosphatases that do the opposite [33]. Among these tyrosine phosphatases, PTP1B has recently been implicated in regulating CAV1-mediated migration processes. Specifically, results from our own group showed that PTP1B controls the migration of melanoma cells by dephosphorylating CAV1 tyrosine-14 [34]. Additionally, this phosphatase is highly sensitive to and inhibited by intracellular ROS levels. Considering that ROS are generated as a byproduct of the metabolic switch, we wondered whether PTP1B may be relevant in controlling migration promoted by metabolic reprogramming. 

To explore this possibility, we first corroborated that PTP1B was present in a complex with CAV1 and found that, indeed, PTP1B co-immunoprecipitated with CAV1 (Figure 6A). To determine if the proximity of these two proteins sufficed to allow PTP1B to control CAV1 tyrosine-14 phosphorylation, we analyzed phosphorylation levels by Western blotting of co-immunoprecipitates in the presence or absence of the specific PTP1B inhibitor Cinn-GEL2-ME. As anticipated, PTP1B inhibition increased CAV1 phosphorylation on tyrosine-14 (Figure 6B). 

In order to determine whether metabolic differences between B16F10(Mock) and B16F10(CAV1) cells were connected to PTP1B, we measured PTP1B activity in PTP1B immunoprecipitates. We observed that PTP1B activity was lower in B16F10(CAV1) than B16F10(Mock) cells and comparable to the levels observed in the presence of the inhibitor (Figure 6C). 

As we showed above, ROS levels were increased by CAV1. To explore the possibility that ROS were acting as endogenous inhibitors of PTP1B activity, we treated B16F10(CAV1) cells with anti-oxidants to decrease ROS levels. We observed that PTP1B activity was restored to levels detected in B16F10 (Mock) cells, confirming that increased ROS production due to CAV1 expression is responsible for controlling PTP1B activity (Figure 6D). As was to be expected, migration and invasion of CAV1-expressing B16F10 (Figure 6E,F), MDA-MB-231 (Figure 6H,I), and HT29(US) (Figure 6K,L) cells increased after PTP1B inhibition. 

Finally, in order to test whether tyrosine-14 phosphorylation affected the role of CAV1 in energy metabolism, we measured lactate production after PTP1B inhibition in mouse melanoma and human breast and colon cancer cell lines. We observed an increase in lactate production upon PTP1B inhibition (Figure 6G,J,M, respectively), indicating that PTP1B regulates cancer cell metabolism by controlling CAV1 phosphorylation. 

### 3.7. Caveolin-1 Induced Metastasis in a Glycolytic Switch-Dependent Manner

Finally, we corroborated the relevance of the CAV1-induced metabolic switch on cancer cell aggressiveness. To that end, we treated B16F10 cells with 2-deoxy-D-glucose (2-DG), a glycolysis inhibitor, and found that 2-DG reduced the migration of CAV1 expressing B16F10(CAV1) cells to levels observed for B16F10(Mock) cells (Figure 7A). To determine the relevance of these observations in vivo, we pre-treated B16F10 cells with 2-DG for 24 h and injected the cells (2 × 10^5^) into the tail vein of C57BL6 mice. After 21 days, the mice were sacrificed, and the black tumor mass in the lungs was visualized (Figure 7B). The percentage of black tumor mass detected indicated that the pre-treatment of B16F10(CAV1) cells with 2-DG for 24 h reduced significantly metastatic nodule formation in the lungs (Figure 7C,D). These data suggest that the CAV1-induced metabolic switch increases glycolysis, which in turn is required to augment the metastatic potential of CAV1-expressing cells.

## 4. Discussion

Cancer cells switch to aerobic glycolysis even in the presence of oxygen, a phenomenon known as the “Warburg Effect,” and do so in order to obtain energy and macromolecular precursors required for proliferation, migration, and invasion (reviewed in [40]). However, many details of the mechanisms involved in such metabolic reprogramming remain to be defined. Our laboratory has recently demonstrated that CAV1 increases migration, invasion, and metastasis of cancer cells lacking E-cadherin in a manner dependent on Y14 phosphorylation and activation of a Rab5/Rac1 signaling axis [37,41]. Given that CAV1 is also ascribed roles related to the regulation of intracellular organelle communication and function (reviewed in [42]), we wondered whether the ability of CAV1 to promote metastasis might also be linked to metabolic changes in such cells. Here, we studied the effect of CAV1 on metastatic cancer cell metabolism and observed that Y14CAV1 phosphorylation induces mitochondrial dysfunction and the Warburg effect, thereby increasing ROS levels, which inhibit PTP1B to augment further pY14-CAV1 levels. In doing so, a feed backward amplification loop is generated that promotes cell migration, invasion, and metastasis (see Figure 8).

The current studies to evaluate the metabolic effect of CAV1 were done in metastatic cells lacking E-cadherin. In fact, E-cadherin, an epithelial cell adhesion protein, plays an important role in determining CAV1 function. Co-expression of CAV1 and E-cadherin reduces tumor formation by sequestering β-catenin in a plasma membrane complex and thereby downregulating β-catenin/Tcf/Lef transcriptional activity [36]. However, on the other hand, expression of CAV1 in cancer cells lacking E-cadherin, which is lost in cancer development during the epithelial to mesenchymal transition [43,44], enhances the metastatic potential by a pathway involving CAV1 phosphorylation on Y14 and activation of the Rab5-Rac1 signaling axis [38,41]. For this reason, we selected metastatic cells lacking E-cadherin to perform the studies reported on here. 

Interestingly, in an in vitro model for epithelial to mesenchymal transition in prostate cancer cells [45] metabolic reprogramming was observed and the expression; additionally, the activity of glycolytic enzymes was higher in mesenchymal cancer than epithelial cells. Others observed in mesenchymal cancer cells that aerobic glycolysis is essential for cytoskeleton remodeling, focal adhesion formation, and production of the energy necessary to migrate [46]. Alternatively, in prostate cancer cells, CAV1 interacts with LRP6 to stimulate the IGF-IR/IR/Akt/mTORC1 pathway and promote glycolysis by increasing the expression of glucose transporters and glycolytic enzymes [16]. Moreover, pY14-CAV1 promotes pseudopodial recruitment of glycolytic enzymes, invadopodia formation, and cell extravasation [47]. Thus, a considerable amount of available data point towards the ability of CAV1/pY14-CAV1 to promote migration and events associated with glycolysis. Consistent with these reports, our results indicate that the expression and phosphorylation of CAV1 on Y14 increase aerobic glycolysis and that they do so by a mechanism that involves shutting down mitochondrial function.

There is controversial data available concerning CAV1 localization and function at the mitochondrial level (reviewed in [42]). Some authors have reported on the presence of CAV1 in mitochondria [48] and others in mitochondria-associated ER membranes [49,50]. Additionally, CAV1 appears necessary for mitochondrial functionality in normal cells [51], but, in cancer cells, mitochondrial CAV1 enhances cell survival in response to stress [52,53] and inhibits mitochondrial respiration increasing tumor malignancy. As mentioned, our results here demonstrate that the elevated expression of CAV1 reduces mitochondrial respiration in cancer cells. In addition, we observed that CAV1 blocks the mitochondrial complex IV, thereby promoting the Warburg effect and the acquisition of more aggressive cancer cell traits. Previous studies described other mechanisms employed by cancer cells to reduce mitochondrial metabolism. For example, aggressive cancer cells express low levels of the mitochondrial pyruvate carrier (MCP), which is responsible for pyruvate entry into the mitochondria. In this way, cancer cells limit mitochondrial pyruvate oxidation and reprogram cell metabolism [54]. Another possibility is the downregulation of AMPK, since mitochondrial AMPK stimulates oxidative phosphorylation, which inhibits aerobic glycolysis and tumor growth. In cancer cells, AMPK expression is reduced to impair mitochondrial metabolism and promote tumor progression [55]. Interestingly, AMPK is required for basal energy production, so the loss of AMPK activity reduces both the glycolytic and mitochondrial metabolism [55]. 

Lactate is a paracrine effector that reprograms the tumor stroma to promote immunosuppression and angiogenesis stimulating cancer progression (reviewed in [56]). On the one hand, lactate serves as a source of energy for stromal carcinoma-associated fibroblasts (CAFs) and mesenchymal stem cells (MSCs) [57], which also provide cancer cells with metabolites [58]. On the other hand, lactate inhibits T and NK cells, promoting an immunosuppressive microenvironment [59]. In addition, lactate stimulates the M2-like polarization of tumor-associated macrophages [60], which are immunosuppressive and enhance cancer cell survival and angiogenesis by secreting vascular endothelial growth factor (VEGF) and other factors together with cytokines [61]. Thus, it is intriguing to speculate that, in doing so, CAV1 not only favors tumor cell metastasis by cell-intrinsic mechanisms as those described above but also by extrinsic pathways involving reprogramming of the tumor stroma. More studies in this area will be required to address this interesting possibility.

ROS control several signaling pathways involved in early (cell transformation and proliferation) and late stages of cancer (invasion, metastasis and angiogenesis). ROS can accumulate inside cells both by endogenous and exogenous mechanisms. Endogenous ROS are produced in subcellular organelles, such as mitochondria and peroxisomes by processes involving cytochrome P-450 or, alternatively, by NADPH oxidases found in many different subcellular locations. Alternatively, pollutants, tobacco smoke, drugs, xenobiotics, and radiation are exogenous sources of ROS (reviewed in [62]). Our results show here that in CAV1-expressing cells, ROS are generated as a bioproduct of mitochondrial dysfunction and then released to the cytosol. As we previously reported, increased CAV1 expression observed in response to the treatment of cancer cells with cytotoxic drugs increases ROS production and the metastatic potential in a manner that is precluded by Src-family kinase inhibitors, underscoring the relevance of tyrosine phosphorylation in this context [39].

The tyrosine phosphatase PTP1B, amongst others, is known to dephosphorylate pY14-CAV1 [33]. Like other tyrosine phosphatases, PTP1B activity depends on a thiol group present in the catalytic pocket, which is reversibly regulated by oxidation; that is, ROS oxidize PTP1B and inhibit its activity [63]. Results from our group showed that stimulation of AT2R in metastatic cancer cells is associated with activation of PTP1B, reduction of pY14-CAV1 levels, Rab5/Rac1 activity, and inhibition of migration, invasion, and metastasis [34]. Consistent with these reports and our results obtained here, we propose that CAV1 promotes a metabolic switch in cancer cells by blocking the mitochondrial complex IV, which induces ROS production and the inhibition of PTP1B. In doing so, pY14-CAV1 levels increase to favor migration and the invasion via Rab5/Rac1 activation in metastatic cancer cells (Figure 8). Importantly, the metabolic switch and concomitant production of ROS increase in the presence of pY14-CAV1 and thereby generate a feed-backward amplification loop. 

Previous studies demonstrated that glycolysis inhibitors synergize with other therapies to reduce cancer-cell aggressiveness [64,65]. Here we demonstrated that 2-DG treatment reduces CAV1-enhanced migration and metastasis, corroborating the relevance of CAV1-induced metabolic reprogramming in determining the metastatic potential of cancer cells. In conclusion, our results demonstrate that CAV1 participates in the metabolic reprogramming of cancer cells by controlling the mitochondrial oxidative phosphorylation pathway. This process is associated with increased lactate and ROS production, as well as migration, invasion, and metastasis. Moreover, CAV1 phosphorylation on Y14 promotes this metabolic shift, as well as the metastatic potential, and is maintained by the inhibition of PTP1B in a ROS-dependent manner. These observations identify CAV1 as a key player in malignant cancer cell behavior and novel potentially interesting targets to control cancer cell metastasis.

## 5. Conclusions

In this study, we evaluated the effect of CAV1 on the metabolic phenotype of cancer cells and observed that expression of CAV1 decreases oxygen consumption in metastatic cells, by blocking mitochondrial complex IV, and induces an increase in ROS at the mitochondrial and cytoplasmic levels. In addition, the expression of CAV1 increases extracellular acidification, indicative of an increase in the glycolytic rate. Taken together, these observations indicate that CAV1 induces a switch from an aerobic oxidative to aerobic glycolytic metabolism, better known as the Warburg effect, which may explain, in part, the increased migratory and invasive rate of metastatic cells expressing CAV1. Interestingly, phosphorylation of CAV1 on Y14 appears to be necessary to induce this CAV1-dependent metabolic switch, which points towards the importance of both CAV1 expression and Y14-phosphorylation for maintenance of the metabolic phenotype in metastatic cancer cells.

## Figures and Tables

**Figure 1 cancers-14-02862-f001:**
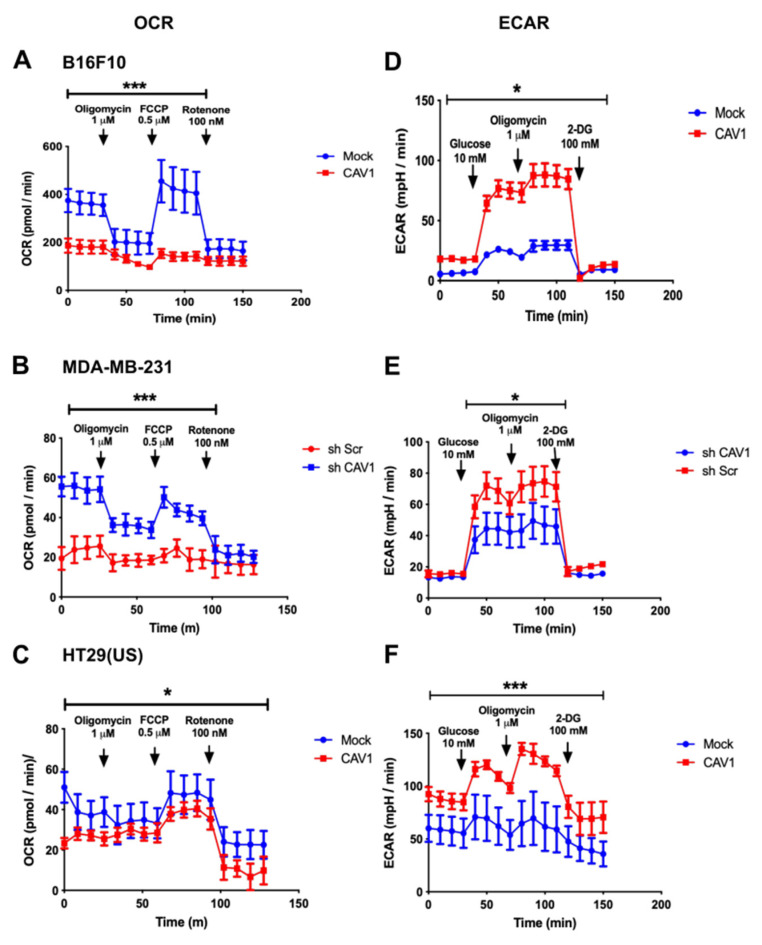
Caveolin-1 expression decreased oxygen consumption and increased the glycolysis rate in metastatic cancer cell lines. (**A**) B16F10 cells (2 × 10^5^), (**B**) MDA-MB-231cells (4 × 10^5^), or (**C**) HT29(US) cells (1 × 10^5^) that express (CAV1 or shScr, red line) or not CAV1 (Mock or shCAV1, blue line) were seeded in a 24-well plate, and 24 h later the oxygen consumption rate (OCR) was measured. To this end, cells were incubated for 1 h with 1 mL per well of Seahorse Bioscience XF24 Calibrant pH 7.4 media supplemented with 100 mM Pyruvate and 5.5 mM glucose at 37 °C. Cells were treated with 1 μM Oligomycin, 0.5 μM Carbonyl cyanide-*p*-trifluoromethoxyphenylhydrazone (FCCP), and finally 100 nM Rotenone. The oxygen consumption rate was evaluated by measuring the changes in the emission of the oxygen fluorophore embedded in the Seahorse sensor plate, using a XF24 Seahorse extracellular flow analyzer. (**D**) B16F10 cells (2 × 10^5^), (**E**) MDA-MB-231 cells (4 × 10^5^), or (**F**) HT29(US) cells (1 × 10^5^) that express (CAV1 or shScr, red line) or not CAV1 (Mock or shCAV1, blue line) were seeded in a 24-well plate, and 24 h later the glycolysis rate was measured by the determination of the extracellular acidification rate (ECAR). Here, cells were incubated for 1 h with 1 mL per well of DMEM media, pH 7.35, supplemented with 143 mM NaCl, 2 mM L-glutamine, and 100 mM 2-Deoxy-D-glucose (2-DG). The extracellular acidification rate was evaluated by measuring the changes in the emission of the proton fluorophore embedded in the Seahorse sensor plate, using a XF24 Seahorse extracellular flow analyzer. The graphs show the averages of results from three independent experiments (mean ± SEM). Significant differences in comparison to the Mock or shCAV1 condition are indicated * *p* < 0.05, *** *p* ≤ 0.001.

**Figure 2 cancers-14-02862-f002:**
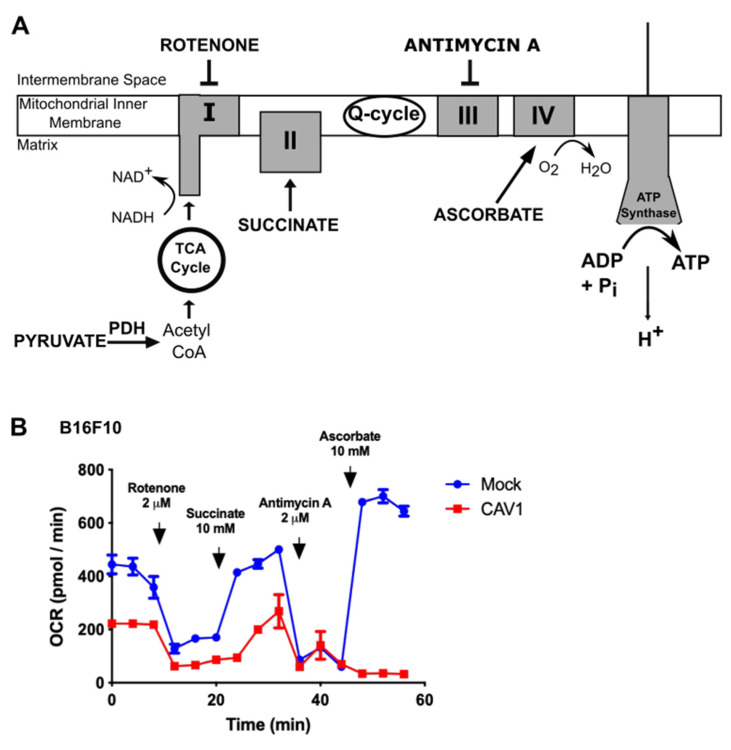
Caveolin-1 expression diminishes oxygen consumption by blocking the mitochondrial IV complex in B16F10 cells. (**A**) B16F10(Mock) or B16F10(CAV1) cells (2 × 10^5^) were seeded in a 24-well plate and 24 h later were permeabilized using 1 nM XF PMP (XF Plasma Membrane Permeabilizer, Seahorse bioscience) plus 4 mM ADP in MAS 1X media supplemented with 10 mM Pyruvate (mitochondrial complex I substrate), 100 nM Rotenone (complex I inhibitor), 10 mM Succinate (complex II substrate), 2 μM antimycin A (complex III inhibitor), and 10 nM Ascorbate plus 100 μM N,N,N’,N’-tetramethyl-*p*-phenylenediamine (TMPD) (complex IV substrate). After each treatment, the oxygen consumption rate (OCR) was measured three times every 4 min. (**B**) The oxygen consumption rate was evaluated by measuring the changes in the emission of the oxygen fluorophore embedded in the Seahorse sensor plate, using a XF24 Seahorse extracellular flow analyzer. The graph shows the average of results from three independent experiments (mean ± SEM) measuring the OCR of B16F10(Mock) (blue line) and B16F10(CAV1) (red line) cells.

**Figure 3 cancers-14-02862-f003:**
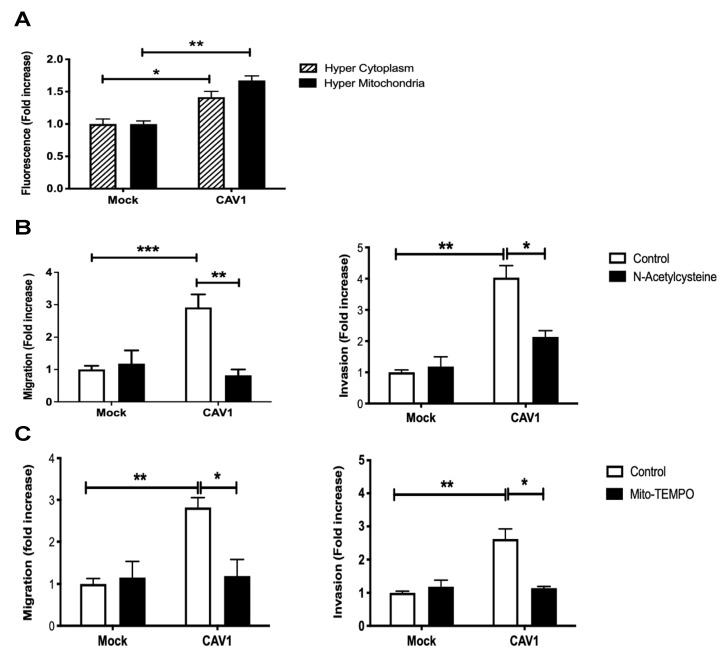
Caveolin-1 expression increased cell migration and invasion in a ROS-dependent manner. (**A**) B16F10(Mock) or B16F10(CAV1) cells (2 × 10^5^) were transfected with 1 μg of pHyPer-dMito (HyPer Mitochondria) or pHyPer-Cyto (HyPer Cytoplasm) plasmids for fluorescent detection of H_2_O_2_. After 24 h, cells were treated for 48 h with 1 mM IPTG. Then, cells were trypsinized, and fluorescence intensity was quantified by flow cytometry. The graphs show the means from three independent experiments (mean ± SEM), whereby the fluorescence intensity in each condition was normalized to the Mock condition. Statistically significant differences are indicated ** *p* < 0.01, * *p* < 0.05. B16F10(Mock) or B16F10(CAV1) cells (6 × 10^5^) were seeded in 6 cm dishes and 24 h later treated with the antioxidants (**B**) N-Acetylcysteine (4 mM for 4 h) or (**C**) Mito-Tempo (5 μM for 2 h). Then, cells (2 × 10^5^) were seeded in (Left panel) Boyden chambers coated with fibronectin (2 μg/mL) and allowed to migrate for 2 h. The cells that migrated through the pores were stained and counted. Values obtained were normalized to the control condition (Mock; on average seven and a half cells per field). Alternatively, cells were seeded (Right panel) in Matrigel-coated chambers and allowed to invade the matrix for 24 h. The cells that accumulated on the lower surface of the membrane were then stained, counted, and normalized to the control condition (Mock; on average four cells per field). The graphs show the averages of results from three independent experiments (mean ± SEM). Statistically significant differences are indicated *** *p* < 0.001, ** *p* < 0.01, * *p* < 0.05.

**Figure 4 cancers-14-02862-f004:**
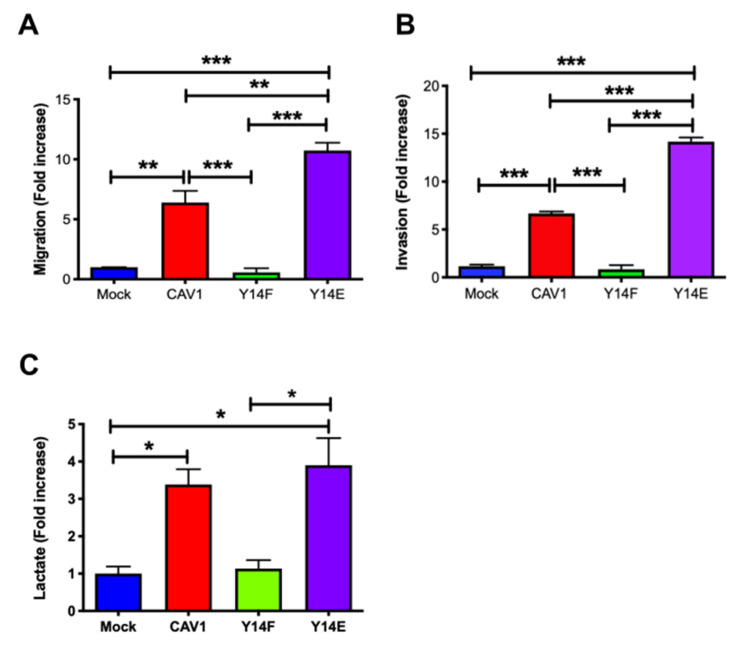
Phosphorylation on tyrosine 14 is necessary for maintenance of the metabolic phenotype induced by the expression of caveolin-1. Cells (2 × 10^5^) B16F10(Mock), B16F10(CAV1), that express non-phosphorylatable CAV1 (B16F10(CAV1-Y14F)) and phosphomimetic CAV1 (B16F10(CAV1-Y14E)) (**A**) were seeded in Boyden chambers coated with fibronectin (2 μg/mL) and allowed to migrate for 2 h. The cells that migrated through the pores were stained and counted. Values obtained were normalized to the control condition (Mock; on average 8 cells per field). (**B**) Cells were seeded in Matrigel-coated chambers and allowed to invade the matrix for 24 h. The cells that accumulated on the lower surface of the membrane were then stained, counted, and normalized to the control condition (Mock; on average 4.1 cells per field). The graphs show the average of results from three independent experiments (mean ± SEM). Significant differences are indicated *** *p* < 0.001, ** *p* < 0.01. (**C**) Cells (6 × 10^5^) were seeded in 6 cm dishes, and 24 h later cells were treated for 48 h with 1 mM IPTG. Then media were collected, and lactate was determined using a L-Lactate Assay kit (Eton Bioscience Inc.) by measuring the absorbance at 490 nm. The graph shows the average of results from three independent experiments (mean ± SEM). The normalized value 1 is equivalent to 2 mM lactate. Significant differences are indicated, * *p* < 0.05.

**Figure 5 cancers-14-02862-f005:**
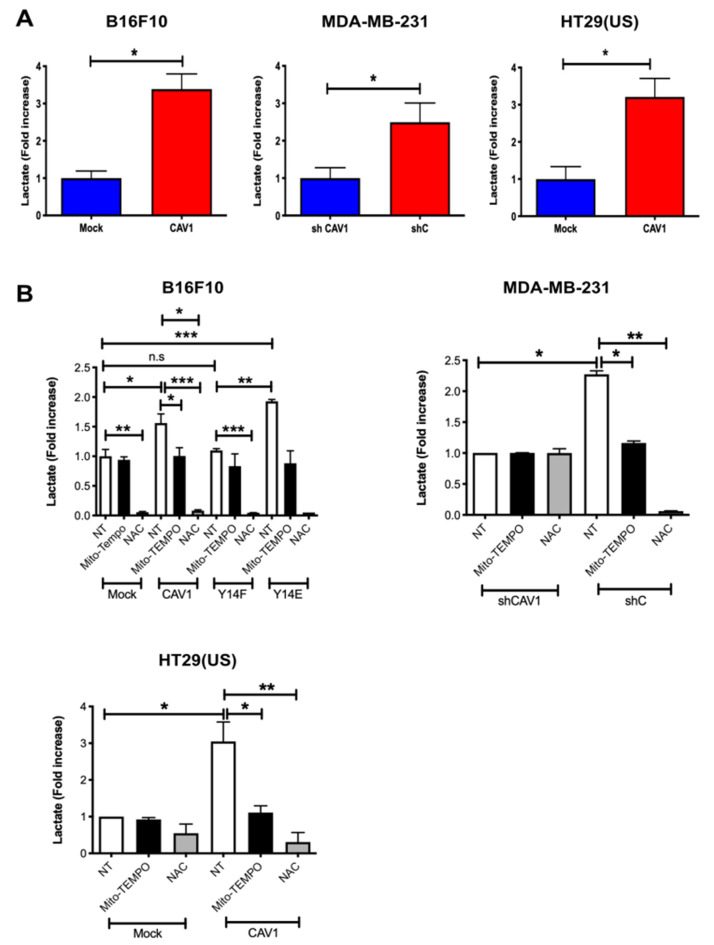
Caveolin-1-induced lactate release is inhibited by the treatment with antioxidants in metastatic cancer cells. (**A**) B16F10(Mock), B16F10(CAV1), MDA-MB-231(shCAV1), MDA-MB-231(shC), HT29(US)(Mock), or HT29(US)(CAV1) and (**B**) B16F10(Mock), B16F10(CAV1), B16F10(CAV1-Y14F), B16F10(CAV1-Y14E), MDA-MB-231(shCAV1), MDA-MB-231(shC), HT29(US)(Mock), or HT29(US)(CAV1) cells (6 × 10^5^) were seeded in 6 cm dishes and 24 h later were treated for 24 h (HT29(US)) or 48 h (B16F10) with 1 mM IPTG. Then, cells were treated with 5 μM Mito-TEMPO for 2 h or 4 mM N-acetylcysteine for 4 h, and media were then collected for lactate measurements at 490 nm. The graph shows the average of results from three independent experiments (mean ± SEM). The normalized value 1 is equivalent to 2 mM lactate. Significant differences are indicated, ns = non-significant, *** *p* < 0.001, ** *p* < 0.01, * *p* < 0.05.

**Figure 6 cancers-14-02862-f006:**
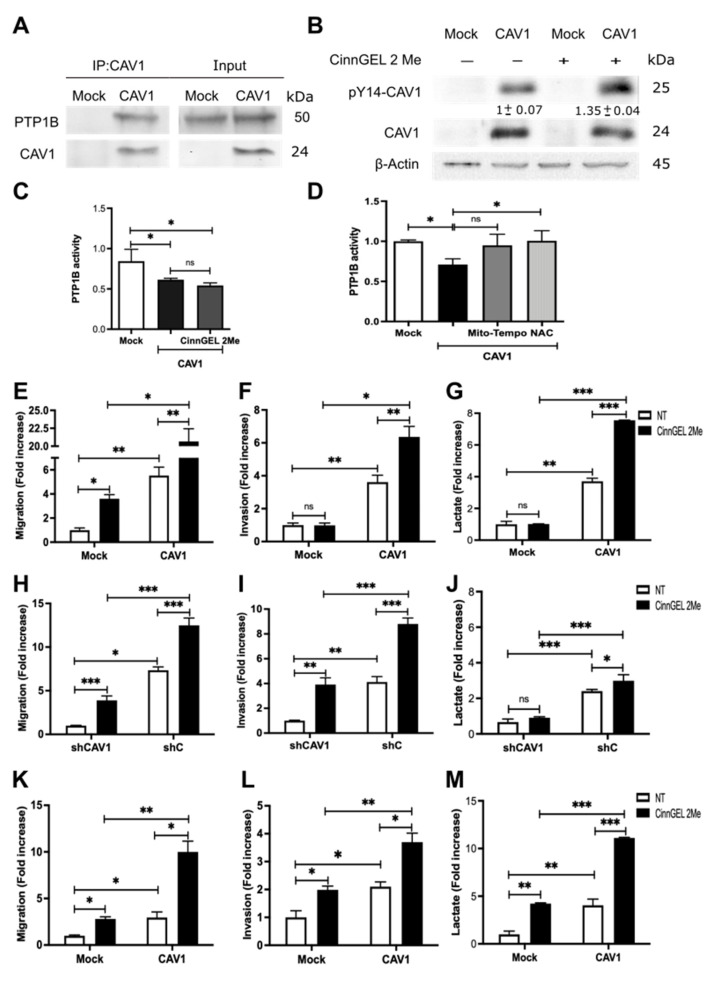
Inhibition of PTP1B activity enhances cell migration, invasion, and lactate production by increasing CAV1 phosphorylation. (**A**) CAV1 was immunoprecipitated from protein extracts containing 2 mg of total protein obtained from B16F10(Mock) and B16F10(CAV1) cells using a polyclonal antibody against CAV1 (2.5 μg per condition). An immunoblot with the total protein input in each experiment is included in the panel to the right. (**B**) B16F10(Mock) and (CAV1) were treated for 12 h with 5 μmol/L CinnGEL 2Me. Total protein extracts were prepared and analyzed by Western blotting with antibodies against CAV1, pY14-CAV1, and β-actin. Western blots representative of results obtained in three independent experiments are shown for A and B. PTP1B was immunoprecipitated from protein extracts containing 2 mg of total protein obtained from B16F10(Mock) and B16F10(CAV1) cells (**C**) treated for 12 h with 5 μmol/L CinnGEL 2Me or (**D**) with 5 μM Mito-TEMPO for 2 h or 4 mM N-acetylcysteine (NAC) for 4 h using a polyclonal antibody against PTP1B (2.5 μg per condition). The immunoprecipitates were resuspended in 50 μL of acetate buffer (pH 5.6), and p-nitrophenyl phosphate was added. Then, the samples were incubated at 37 °C, and absorbance was measured at 415 nm. B16F10(Mock), B16F10(CAV1), MDA-MB-231(shCAV1), MDA-MB-231(shC), HT29(US)(Mock), and HT29(US)(CAV1) were treated for 12 h with 5 μmol/L CinnGEL 2Me. Then, (**E**) B16F10 cells (2 × 10^5^) (**H**) MDA-MB-231 or (**K**) HT29(US) were seeded in Boyden chambers coated with fibronectin (2 μg/mL) and allowed to migrate for 2 h. The cells that migrated through the pores were stained and counted. Alternatively, (**F**) B16F10, (**I**) MDA-MB-231, or (**L**) HT29(US) cells were seeded in Matrigel-coated chambers and allowed to invade the matrix for 24 h. Values obtained were normalized to the control condition. Then, the media were collected, and lactate was measured at 490 nm for (**G**) B16F10, (**J**) MDA-MB-231, or (**M**) HT29(US) cells. The normalized value 1 is equivalent to 3.57 mM lactate. The graphs show the average of results from three independent experiments (mean ± SEM). Significant differences are indicated, ns = non-significant, *** *p* < 0.001, ** *p* < 0.01, * *p* < 0.05.

**Figure 7 cancers-14-02862-f007:**
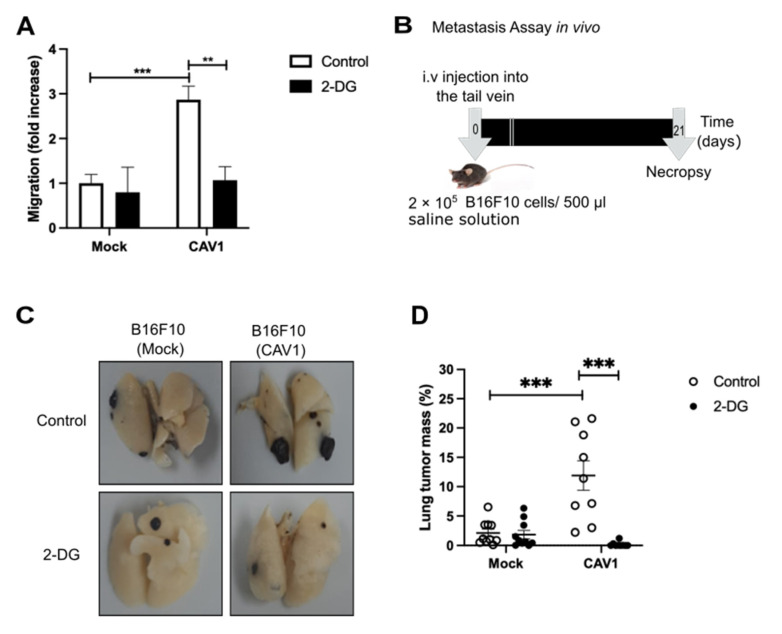
Inhibition of glycolysis reduces migration and metastasis. B16F10(Mock) and B16F10(CAV1) cells were treated for 24 h with 2-Deoxy-D-glucose (2-DG, 1 mM). (**A**) Cells (2 × 10^5^) were seeded in Boyden chambers coated with fibronectin (2 μg/mL) and allowed to migrate for 2 h. The cells that migrated through the pores were stained and counted. Values obtained were normalized to the control condition (Mock; on average 18 cells per field). (**B**) Schematic representation of the experimental design used for the metastasis assay. B16F10(Mock) and B16F10(CAV1) cells (2 × 10^5^) were pre-treated for 24 h with 1 mM 2-DG and injected intravenously into the tail vein of C57BL/6 mice. After 21 days, mice were euthanized, lungs were fixed, and total lung and metastasis mass were determined. (**C**) Representative images of the lung metastasis in each condition are shown. (**D**) Average values (mean ± SEM) for lung occupation by tumors in C57BL/6 mice inoculated with B16F10(Mock) and B16F10(CAV1) cells, pre-incubated or not with 2-DG. Statistically significant differences are indicated (*** *p* < 0.001, ** *p* < 0.01; *n* = 10 animals per condition).

**Figure 8 cancers-14-02862-f008:**
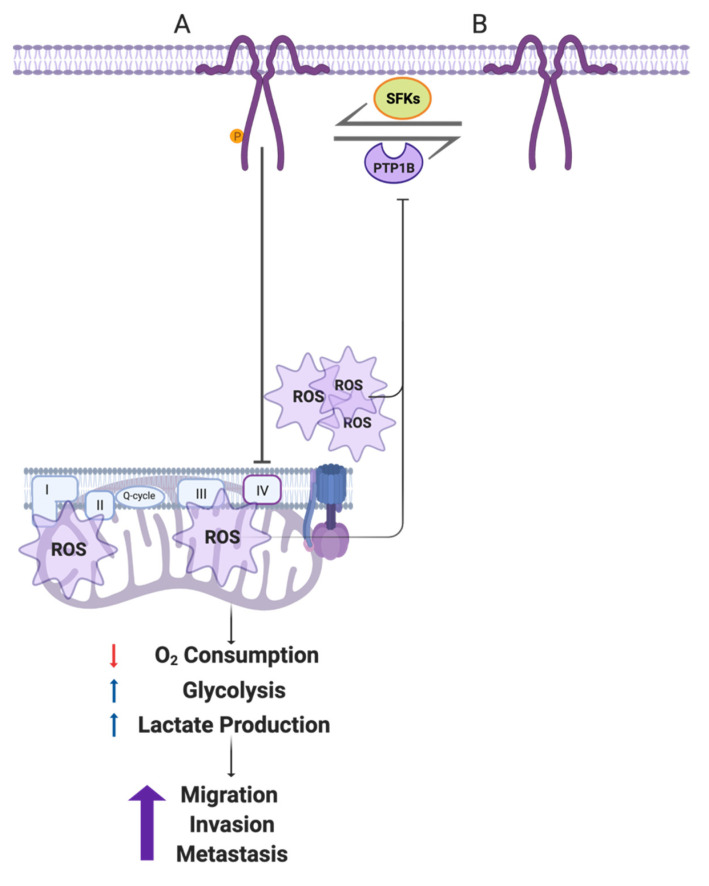
Working model summarizing the main findings described in this study. (**A**) In metastatic cells, CAV1 is phosphorylated on Y14 by Src family kinases. pY14-CAV1 inhibits the mitochondrial IV complex or impairs the ability of the cell to recover from complex III damage, increasing the ROS production (shown as emanating from complex III, as well as complex I/II) and accumulation both in mitochondria and cytosol. Mitochondrial inhibition decreases the oxygen consumption and promotes a metabolic Warburg-like switch to an aerobic glycolytic phenotype. These events enhance cell migration, invasion, and metastasis driven by CAV1 expression. In addition, CAV1 expression and phosphorylation on Y14 inhibits the activity of the phosphatase PTP1B (**B**) by augmenting ROS. In this way, the metabolic switch and ROS formation generate a feed-backward amplification loop to promote CAV1 phosphorylation on Y14 and thereby the increase in cell migration, invasion, and metastasis. Created with BioRender.com (Accessed on 5 May 2022).

**Table 1 cancers-14-02862-t001:** Reagents used for the preparation of the MAS test medium.

Reagent	MAS 3 × Media Concentration
Mannitol	660 mM
Saccharose	210 mM
KH_2_PO_4_	30 mM
HEPES	6 mM
EGTA	3 mM
Fatty Acid Free BSA	0.6% (*m*/*v*)

## Data Availability

The data presented in this study are available in this article.

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
