# Peer review of "Mitochondrial Dysfunction and the Glycolytic Switch Induced by Caveolin-1 Phosphorylation Promote Cancer Cell Migration, Invasion, and Metastasis"

_cancers, 2022, doi:10.3390/cancers14122862_

Round 1

Reviewer 1 Report

A very interesting study showing that tyrosine phosphorylated caveolin-1 (pCav1) reduces mitochondrial respiration and promoted tumor cell migration and invasion via ROS production and increased glycolysis. This was shown to be mediated by ROS inhibition of the PTPB1 phosphatase increasing pCav1 levels. Using multiple cancer cell lines, ROS and PTP1B regulation of migration, invasion and lactate production are shown to be Cav1-dependent.  Data showing that PTPB1 inhibition increases ROS and that the increased migration and invasion due to PTPB1 inhibition are ROS-dependent would strengthen the paper.

The recent Jiang et al Redox Biology 52:102304 paper from the Minshall group should be cited as should the review on pCAV in cancer from the Nabi lab (Wong et al, Cancer Met Rev. 39(2):455-469)

Author Response

Thank you so much for your positive feedback and comments, here is our response 

Reviewer 1:

Reviewer comment

A very interesting study showing that tyrosine phosphorylated caveolin-1 (pCav1) reduces mitochondrial respiration and promoted tumor cell migration and invasion via ROS production and increased glycolysis. This was shown to be mediated by ROS inhibition of the PTPB1 phosphatase increasing pCav1 levels. Using multiple cancer cell lines, ROS and PTP1B regulation of migration, invasion and lactate production are shown to be Cav1-dependent.  Data showing that PTPB1 inhibition increases ROS and that the increased migration and invasion due to PTPB1 inhibition are ROS-dependent would strengthen the paper.

Author response: We that this reviewer for the very positive evaluation of our manuscript.

Reviewer comment:

The recent Jiang et al Redox Biology 52:102304 paper from the Minshall group should be cited as should the review on pCAV in cancer from the Nabi lab (Wong et al, Cancer Met Rev. 39(2):455-469)

Author response: We have read the indicated papers and now incorporated both into our manuscript, as the references number 53 and 26, respectively.

Reviewer 2 Report

In this study, the authors demonstrate an effect of phosphorylated caveolin-1 on cancer migration, invasion and metastasis via an inhibition of complex IV of respiratory chain and an activation of glycolysis.

Do the author have an idea on the mechanism of complex IV inhibition by caveolin-1 ?

I have only minor comments on this article:

P 2 line 58 “….decreased expression of CAV1….” and line 69 “….CAV1 expression….” Seem contradictory.

P 6-7 line 277. I see an increase and not a decrease in basal glycolysis when CAV1 is expressed (red line) in Fig. F.

p 7 Figure 1, perhaps put OCR and ECAR at the top of the corresponding columns to facilitate the reading of the figure.

P 8 line 315-316: I wouldn’t write: “this inhibition was reverted…” because the inhibition of complex III still occurs, but perhaps “this inhibition is bypassed by…”.

P8 line 316: “….ascorbate and TMPD 100, two complex IV substrates…”. Ascorbate and TMPD are not two (independent) substrates of complex IV, but constitute a system to reduce cytochrome C, the actual substrate of complex IV. See for instance: David G. Nicholls, Stuart Ferguson Bioenergetics, Fourth Edition p 58: “ For example, the redox dye TMPD can shuttle electrons from external ascorbate to cyt c, allowing complex IV to be studied by itself in the intact mitochondrion.”. I could write: “….ascorbate and TMPD 100, to reduce cytochrome C that deliver electrons to cytochrome C oxidase…”.

P 11 legend of Fig. 4: the definition of Y14F and Y14E should be repeated in the legend to have a complete and independent understanding of the figure.

P 17 line 547: [reviewed in [38]]. Isn’t it rather [37]? Check reference numbering. Idem line 585. Shouldn’t [40] rather be [39]?

P 19: this figure is not very clear. CAV1 should be indicated (written). ROS appear to originate from complex IV and ATP synthase (to be moved to complex I and III and possibly II). The feed-backward amplification loop is not clear on the figure.

Author Response

Thank you so much for your positive feedback and comments, here is our response 

Reviewer 2:

Reviewer comment:

In this study, the authors demonstrate an effect of phosphorylated caveolin-1 on cancer migration, invasion and metastasis via an inhibition of complex IV of respiratory chain and an activation of glycolysis.

Do the author have an idea on the mechanism of complex IV inhibition by caveolin-1?

Author response: This is of course a very interesting question. To shed light on this, we have characterized the proteins that co-immunoprecipitated with caveolin-1 in B16F10(CAV1) cells. These were initially trypsin digested and the resulting peptides were analyzed by mass spectrometry. The identified peptide fragments were then compared using the database Mascot. According to those results, the mitochondrial atypical kinase COQ8A co-immunoprecipitated with CAV1. This kinase participates in the biosynthesis of coenzyme Q (Stefely  J, Reidenbach A, Ulbrich A, Oruganty K, et al. Mitochondrial ADCK3 Employs an Atypical Protein Kinase-like Fold to Enable Coenzyme Q Biosynthesis. Molecular Cell. 2015;57(1):83-94.) and association with CAV1 may reduce its antioxidant activity, which would explain why antioxidants reduce CAV1-induced cell migration and invasion.

Please note that this additional information was not included in the manuscript because the studies are still at a rather preliminary stage. However, if this reviewer deems it important to share the available information with the readership, for instance in the discussion, we would be happy to do so.

Reviewer comments with author responses:

I have only minor comments on this article:

Comment: P 2 line 58 “….decreased expression of CAV1….” and line 69 “….CAV1 expression….” Seem contradictory.

Response: we agree. The text has been modified to specify in the first case (line 58) that the decrease in CAV1 expression has been reported in cancer-associated fibroblasts (CAFs). On the other hand, the expression of CAV1 refers to cancer cells (line 69).

Comment: P 6-7 line 277. I see an increase and not a decrease in basal glycolysis when CAV1 is expressed (red line) in Fig. F.

Response: Indeed this is the case and the error has been corrected.

Comment: p 7 Figure 1, perhaps put OCR and ECAR at the top of the corresponding columns to facilitate the reading of the figure.

Response: The figure has been modified as suggested.

Comment: P 8 line 315-316: I wouldn’t write: “this inhibition was reverted…” because the inhibition of complex III still occurs, but perhaps “this inhibition is bypassed by…”

Response: The text has been modified as suggested and now reads as follows: … and this inhibition was bypassed by addition of ascorbate and TMPD 100, that deliver electrons to and reduce the cytochrome C oxidase, a substrate of complex IV (Figure 2A).  

Comment: P8 line 316: “….ascorbate and TMPD 100, two complex IV substrates…”. Ascorbate and TMPD are not two (independent) substrates of complex IV, but constitute a system to reduce cytochrome C, the actual substrate of complex IV. See for instance: David G. Nicholls, Stuart Ferguson Bioenergetics, Fourth Edition p 58: “ For example, the redox dye TMPD can shuttle electrons from external ascorbate to cyt c, allowing complex IV to be studied by itself in the intact mitochondrion.”. I could write: “….ascorbate and TMPD 100, to reduce cytochrome C that deliver electrons to cytochrome C oxidase…”.

Response: The text has been modified as suggested and now reads as follows: … and this inhibition was bypassed by addition of ascorbate and TMPD 100, that deliver electrons to and reduce the cytochrome C oxidase, a substrate of complex IV (Figure 2A).  

Comment: P 11 legend of Fig. 4: the definition of Y14F and Y14E should be repeated in the legend to have a complete and independent understanding of the figure.

Response: As requested the definition of the cells employed has been included in the legend to Figure 4.

Comment: P 17 line 547: [reviewed in [38]]. Isn’t it rather [37]? Check reference numbering. Idem line 585. Shouldn’t [40] rather be [39]?

Response: We had a problem with the program Endnote. The references have been revised and corrected as indicated.

Comment: P 19: this figure is not very clear. CAV1 should be indicated (written). ROS appear to originate from complex IV and ATP synthase (to be moved to complex I and III and possibly II). The feed-backward amplification loop is not clear on the figure.

Response: We have modified the figure to indicate more clearly how the feed-backwards loop is generated. Also the mitochondrial ROS are depicted as emanating from the complexes I/II and III as suggested.